# Tailoring 3HV Fraction in Poly(3-hydroxybutyrate-*co*-3-hydroxyvalerate) by *Azotobacter vinelandii* Through Oxygen and Carbon Limitation in Continuous Cultures

**DOI:** 10.3390/polym17192578

**Published:** 2025-09-24

**Authors:** Andrés Pérez, Andrés García, Viviana Urtuvia, Carlos Peña, Alvaro Díaz-Barrera

**Affiliations:** 1Escuela de Ingeniería Bioquímica, Pontificia Universidad Católica de Valparaíso, Valparaíso 2340025, Chile; aaperezb17@gmail.com (A.P.); viviana.urtuvia.gatica@gmail.com (V.U.); 2Centro de Investigación en Biotecnología, Universidad Autónoma del Estado de Morelos, Cuernavaca 62210, Morelos, Mexico; andres.garcia@docentes.uaem.edu.mx; 3Departamento de Ingeniería Celular y Biocatálisis, Universidad Nacional Autónoma de México, Cuernavaca 62210, Morelos, Mexico; carlos.pena@ibt.unam.mx

**Keywords:** Poly(3-hydroxybutyrate-*co*-3-hydroxyvalerate), 3HV fraction, agitation rates, *Azotobacter vinelandii* OP, specific oxygen uptake rate

## Abstract

*Azotobacter vinelandii* OP is a bacterium that can produce poly(3-hydroxybutyrate-*co*-3-hydroxyvalerate) (P3HBV), a biodegradable and biocompatible polymer with applications in the biomedical field. This study aimed to evaluate P3HBV production and its 3-hydroxyvalerate (3HV) fraction under different agitation rates and oxygen uptake rates (q_O2_) in chemostat cultures of *A. vinelandii* OP. Steady-state conditions with either oxygen or carbon limitation were established by modulating the agitation rates. Under oxygen-limited conditions (low q_O2_ values) biomass and P3HBV concentrations increased to 3.3 g L^−1^ and 2.1 g L^−1^, respectively. At higher q_O2_ values, the chemostat cultures were limited by carbon, and P3HBV content decreased from 62% to 33% (w w^−1^). The highest 3HV molar fractions, 33.7 and 36.4 mol %, were observed at both the lowest and highest q_O2_ levels, possibly linked to comparable valeric acid consumption rates. An elevated NAD(P)H/NAD(P)^+^ ratio was also observed under oxygen limitation, favoring polymer accumulation by indicating a more favorable intracellular redox state. These findings highlight the impact of nutrient limitation and respiratory activity on the biosynthesis of P3HBV and the 3HV composition by *Azotobacter vinelandii* OP. Such insights can support the development of tailored bioprocesses to modulate polymer characteristics, enabling a broader range of potential biomedical applications for P3HBV.

## 1. Introduction

Polyhydroxyalkanoates (PHAs) are polymers composed of ester-linked monomers and are widely used in biomedical applications [1,2]. Microorganisms can produce these polymers by converting diverse carbon sources, such as sugars, oils, organics acids, alcohols, and waste streams, into intracellular polyesters [3]. PHAs also play a role in microbial response to oxidative stress and serve as energy storage compounds [4,5]. They are biodegradable, biocompatible, and can be synthesized from renewable substrates, which makes them a sustainable alternative to petrochemical plastic, with applications ranging from packaging to high-value biomedical uses [1,6]. Poly-3-hydroxybutyrate (P3HB) is the most common homopolymer in the PHA family [6]. However, its high hydrophobicity, brittle mechanical properties, and slow biodegradation rate limit its application, particularly in the biomedical and pharmaceutical fields. A promising approach to enhance its properties is the incorporation of monomers with varying chain lengths into the polymer chain [7]. The addition of 3-hydroxyvalerate (3HV) results in a copolymer, poly(3-hydroxybutyrate-*co*-3-hydroxyvalerate) (P3HBV), which exhibits improved thermomechanical properties, and it is both biodegradable and biocompatible; therefore, it is suitable for biomedical applications [1,8].

The 3HV fraction strongly influences P3HBV properties and enables customization for high-value applications [9,10]. Low 3HV contents (3–10 mol %) reduce brittleness, making P3HBV suitable for rigid biomedical devices and bioresorbable packaging [11]. Intermediate levels (10–20 mol %) decrease crystallinity and enhance elasticity, favoring uses in tissue scaffolds, drug delivery, and flexible films [12]. Higher fractions (20–30 mol %) improve elongation and flexibility, ideal for implantable devices and regenerative medicine [12]. Though very high 3HV contents (up to 60 mol %) have been reported, they often compromise mechanical strength. Therefore, the 3HV fraction should be tailored to meet the specific mechanical and degradation requirements of each application [1,7,8].

The production of P3HB and related PHAs has been described in more than 300 bacterial species, including *Azotobacter vinelandii,* which is capable of fixing atmospheric nitrogen and accumulating more than 80 % of its dry cell weight in the form of biopolymer [13,14,15]. *A. vinelandii* OP strain is particularly suitable as a model because it is non-pathogenic, exhibits fast growth, and synthesizes high levels of PHA from diverse carbon sources, including organic acids such as valerate [7]. This versatility allows for the efficient production of P3HBV and makes *A. vinelandii* OP an attractive alternative to other PHA-producing bacteria. However, most studies with *A. vinelandii* OP have been restricted to batch and fed-batch systems, and its potential in continuous cultivation remains largely unexplored [7]. In addition, previous reports have generally evaluated only one or two agitation rates, without performing a systematic analysis across a wide range of agitation rates. This gap limits our understanding of how this bacterium behaves under steady-state conditions and how operational parameters such as agitation influence copolymer composition.

The biosynthesis of short-chain-length PHAs (scl-PHAs) such as P3HB involves three key enzymatic steps: condensation of two acetyl-CoA molecules by β-ketothiolase (*phb*A), reduction of acetoacetyl-CoA to (R)-3-hydroxybutyryl-CoA by NADPH-dependent reductase (*phb*B), and polymerization by PHA synthase (*phb*C) [16,17]. When fatty acids are utilized as carbon sources, the β-oxidation pathway produces intermediates such as (R)-3-hydroxyacyl-CoA, formed through the catalytic action of trans-enoyl-CoA hydratase (PhaJ) [18,19,20,21]. In the presence of valeric acid, *A. vinelandii* synthesizes the copolymer P3HBV through this pathway [22,23,24,25]. While most studies on PHA copolymerization have been performed in batch or fed-batch cultures [26], limited work has addressed P3HBV production under continuous conditions. In particular, there are no systematic studies linking agitation rate and oxygen transfer with 3HV incorporation in *A. vinelandii* chemostat cultures, despite the known importance of redox balance in modulating copolymer composition [17,18,27,28].

The agitation rate and, therefore, the oxygen transfer rate (OTR) have important effects on the modulation of PHA copolymers [18]. Previous studies have demonstrated that, in bioreactor cultures of *A. vinelandii* OP, values of OTR_max_ of 17.2 ± 1.2 mmol L^−1^ h^−1^ lead to an increase in the molar fraction of 3HV (34.9 ± 6.6 mol %) in the P3HBV copolymer chain, compared with low OTR_max_ conditions (4.3 ± 0.7 mmol L^−1^ h^−1^), where a 3HV content of 18.5 ± 6.4 mol % was obtained [25]. Although most research on PHA copolymerization has been conducted in batch cultures, studies focused on P3HBV production under single-stage continuous cultivation remain limited.

Continuous production strategies, such as chemostat cultivation, offer several advantages, including constant polymer composition, better control over process parameters, and potentially higher productivity [29,30,31]. The continuous mode enables the use of inhibitory organics acids, such as valeric or propionic acid, which have been observed to accumulate at relatively high concentrations in batch cultures. However, these organics acids can be maintained at subinhibitory levels in the culture medium, ensuring more stable and reproducible P3HBV synthesis [19,30,32].

In chemostat cultures with *Alcaligenes eutrophus*, P3HBV production is modulated by controlling the concentration of propionic or pentanoic acids, resulting in HV contents between 5 and 38 mol %, depending on the carbon substrate used [32]. As shown by Yu et al. [33], it is possible to regulate the 3HV content of P3HBV copolymers in single-stage chemostat cultures of *Cupriavidus necator* by adjusting the ratio of glucose to sodium propionate in the feed. They reported 3HV contents ranging from 2 to 60 mol % and noted that higher propionate concentrations were correlated with increased 3HV incorporation.

They also reported that a lower dilution rate (D) (0.05 h^−1^) promoted a higher polymer accumulation and stability during steady-state operation [29]. These results highlight the effectiveness of continuous cultivation for controlling the polymer composition and enhancing the productivity of P3HBV biosynthesis. Furthermore, continuous culture allows for the tuning of polymer properties by adjusting the feed composition, resulting in more uniform copolymers with consistent properties.

However, the use of chemostat systems for producing P3HBV using *A. vinelandii* OP has not been extensively investigated. A significant literature gap exists regarding its behavior in continuous cultures, especially under varying agitation rates that influence the OTR and copolymer composition. Unlike batch and fed-batch cultures, chemostat cultivation allows the specific growth rate (µ) to be kept constant (µ = D), providing stable metabolic conditions that enable more precise control of polymer accumulation and composition [19,30]. This makes continuous cultivation a powerful approach to systematically evaluating how agitation and oxygen transfer modulate 3HV incorporation in *A. vinelandii* OP. Therefore, the present study aimed to assess the relationship between the oxygen uptake rate and the 3HV composition of the P3HBV copolymer produced by *A. vinelandii* OP in chemostat cultures.

## 2. Materials and Methods

### 2.1. Bacterial Strain, Culture Medium, and Inoculum Preparation

*A. vinelandii* OP (ATCC 13705) was grown under nitrogen-fixing conditions in a culture medium with the following composition (g L^−1^): 20 sucrose, 0.66 K_2_HPO_4_, 0.16 KH_2_PO_4_, 0.056 CaSO_4_·2H_2_O, 0.2 NaCl, 0.2 MgSO_4_·7H_2_O, 0.0029 Na_2_MoO_4_·2H_2_O, and 0.027 FeSO_4_·7H_2_O. Sucrose, K_2_HPO_4_, and KH_2_PO_4_ were dissolved in a shake flask, bioreactor, or feed tank and sterilized by autoclaving at 121 °C for 20 min. Calcium and mineral salt solutions were prepared in separate bottles and autoclaved. The composition of the culture medium in the feed tank was the same, with the addition of valeric acid (1 g L^−1^) [21] and a volume of 15 L. The inoculum was prepared in a 500 mL Erlenmeyer flask with 100 mL of culture medium and incubated at 30 °C and 200 rpm in an orbital shaker (Daihan LabTech Co., Ltd. Namyangju, Kyungki-Do, Republic of Korea). After 24 h, the bioreactor was inoculated with 10% (v v^−1^) of this inoculum.

### 2.2. Chemostat Cultures

Chemostat cultures were performed in a 3 L bioreactor (Applikon, Schiedam, Netherlands; 1.5 L working volume). pH was controlled at 7.1 ± 0.1 with 2 N NaOH (Masterflex^®^ pump coupled to Alpha 800 pH/ORP Controller & Transmitter). The reactor was equipped with two Rushton turbines, aerated at 1.5 L min^−1^, and operated at agitation rates of 250–850 rpm. DOT was monitored with a polarographic probe (Ingold, Mettler-Toledo, Melbourne, Australia). After 24 h, fresh medium was fed, and culture broth was continuously withdrawn using a peristaltic pump (Cole-Parmer, Vernon Hills, IL, USA). The chemostat cultures were operated at a D of 0.04 h^−1^, corresponding to one-third of the specific growth rate (μ = 0.12 h^−1^) previously obtained in batch cultures. This low dilution rate was selected to ensure a longer residence time and promote P3HBV accumulation [33]. The steady-state condition was achieved after three residence times, and the biomass concentration remained constant (≤10% variation). For the analysis culture samples (50 mL) were taken from the bioreactor for analysis. All experiments were performed in triplicate, and results are reported as mean values.

### 2.3. Analytical Methods

Biomass was determined gravimetrically. Culture broth (50 mL) was centrifuged at 10,000 rpm for 10 min (Thermo Scientific SL-16R, Waltham, MA, USA), and the pellet was washed three times with distilled water before drying at 105 °C to constant weight.

Sucrose and valeric acid were quantified by HPLC (Jasco LC-4000 series, Hachioji, Tokyo, Japan) using an Aminex HPX-87H ion-exclusion column (300 × 7.8 mm, Bio-Rad, Hercules, CA, USA) coupled to RI (RI-4030, Jasco Corp., Hachioji, Tokyo, Japan) and UV (UV-4030, Jasco Corp., Hachioji, Tokyo, Japan) detectors. A volume of 20 µL was injected, and 1.5 mM H_2_SO_4_ was used as the mobile phase (0.6 mL min^−1^, 55 °C, 50 min). Standard curves were prepared with sucrose (≤20 g L^−1^) and valeric acid (≤5 g L^−1^) [25].

For P3HBV determination and monomeric composition, the samples were analyzed by GC (PerkinElmer Clarus 600, PerkinElmer Inc., Waltham, MA, USA) equipped with an FID (Clarus 600S, PerkinElmer Inc., Waltham, MA, USA) [34]. Approximately 10 mg dry biomass was subjected to propanolysis, and propyl ester extracts were injected into an Elite-5 MS capillary column (30 m × 0.25 mm × 0.25 µm, PerkinElmerInc., Waltham, MA, USA) in split mode (1:50) using helium (0.71 mL min^−1^) as carrier gas. Benzoic acid and P3HBV (Sigma-Aldrich, Darmstadt, Germany) served as internal and external standards, respectively [25,35,36].

### 2.4. Determination of the OTR, CTR, and Estimation of the Specific Oxygen Uptake Rate and RQ

The oxygen transfer rate (OTR) and carbon dioxide transfer rate (CTR) were determined during chemostat transients and at each steady state by online measurement of oxygen and carbon dioxide in the exhaust gas using a gas analyzer (Teledyne Instruments, Model 7500, Thousand Oaks, CA, USA). OTR and CTR were calculated from gas mass balances according to Equations (1) and (2), respectively [37].(1)$$OTR=\frac{F_{G}^{in}}{V_{R}V_{M}}\left(X_{O_{2}}^{in}-X_{O_{2}}^{out}\left(\frac{1-X_{O_{2}}^{in}-X_{{CO}_{2}}^{in}}{1-X_{O_{2}}^{out}-X_{{CO}_{2}}^{out}}\right)\right)$$(2)$$CTR=\frac{F_{G}^{in}}{V_{R}V_{M}}\left(X_{{CO}_{2}}^{in}\left(\frac{1-X_{O_{2}}^{in}-X_{{CO}_{2}}^{in}}{1-X_{O_{2}}^{out}-X_{{CO}_{2}}^{out}}\right)-X_{{CO}_{2}}^{out}\right)$$

OTR and CTR are expressed in mmol L^−1^ h^−1^, the parameter $F_{G}^{in}$ represents the volumetric inlet air flow under standard conditions (L h^−1^); V_R_ corresponds to the working volume (L); V_M_ is the molar volume of the ideal gas under standard conditions (L mmol^−1^); $X_{{CO}_{2}}^{in}$ is the molar fraction of oxygen in the inlet air (mol mol^−1^); $X_{O_{2}}^{out}$ is the molar fraction of oxygen in the outlet fermentation gas of the bioreactor (mol mol^−1^); $X_{{CO}_{2}}^{in}$ is the molar fraction of carbon dioxide in the inlet air (mol mol^−1^); and $X_{{CO}_{2}}^{in}$ is the molar fraction of carbon dioxide in the outlet fermentation gas of the bioreactor (mol mol^−1^).

The specific oxygen uptake rate (q_O2_) (mmol g^−1^ h^−1^) (Equation (3)) and the respiratory quotient (RQ) (Equation (4)) were calculated in each steady state according to the following:(3)$$q_{O_{2}}=\frac{OTR}{X\prime }$$(4)$$RQ=\frac{CTR}{OTR}$$
where X′ is the mean value of biomass without including P3HBV (g L^−1^) obtained in each steady state.

### 2.5. Measurements of the Intracellular NAD^+^, NADH, NADP^+^, and NADPH Concentrations

The levels of NAD^+^, NADH, NADP^+^, and NADPH were determined using enzymatic methods [38]. This analysis reflects the intracellular redox state linked to respiration and P3HBV metabolism. Cofactors were extracted and quantified with the EnzyChrom™ assay kit following the manufacturer’s instructions (BioAssay Systems, Hayward, CA, USA). Briefly, 10 mL of culture broth was immediately mixed with cold 70% (v v^−1^) methanol to rapidly inactivate cellular metabolism [39]. The resulting cell pellet was washed with cold PBS and resuspended in the cofactor-specific extraction buffer provided by the kit to isolate the reduced or oxidized pyridine nucleotides. NAD(H) and NADP(H) levels were measured using glucose-6-phosphate dehydrogenase and lactate dehydrogenase, respectively, with absorbance read at 565 nm on a Thermo Scientific Multiskan™ GO microplate spectrophotometer.

### 2.6. Carbon Balance

The carbon distribution was determined based on mass balances in the different steady states evaluated (Equation (5)).(5)$${\%C}_{Recovered}={\%C}_{Biomass}+{\%C}_{P3HBV}+{\%C}_{{CO}_{2}}$$
where ${\%C}_{Recovered}$ is the sum total of carbon used. ${\%C}_{Biomass}$ is the carbon fraction used for biomass, was estimated for biomass-free P3HBV. ${\%C}_{P3HBV}$ is the carbon fraction for P3HBV, the 3HB and 3HV fractions were summed, where sucrose contributes to 3HB and valeric acid to 3HV. ${\%C}_{{CO}_{2}}$ is the carbon fraction directed to CO_2_, and it was calculated considering the CTR (mmol L^−1^ h^−1^) measurements.

The sum of the carbon from sucrose and valeric acid was considered the total carbon, which is mainly distributed into biomass, P3HBV, and CO_2_.

### 2.7. Estimation of Fermentation Parameters

Biomass yield based on sucrose (Y_X/S_), the yield of P3HBV based on sucrose (Y_P3HBV/S_), the volumetric productivity (Q_P3HBV_), specific sucrose consumption rate (q_S_), and the specific valeric uptake rate (q_Val_) were determined in each steady state according to the following Equations:(6)$$Y_{X/S}=\frac{X}{S_{R}-S}$$(7)$$Y_{P3HBV/S}=\frac{P}{S_{R}-S}$$(8)$$q_{S}=\left(\frac{S_{R}-S}{X\prime }\right)D$$(9)$$q_{Val}=\left(\frac{{Val}_{R}-Val}{X\prime }\right)D$$(10)$$Q_{P3HBV}=DP$$
where X is the mean value of the biomass including the P3HBV content (g L^−1^), X′ represents the mean value of the biomass excluding the P3HBV content (g L^−1^), and P is the mean P3HBV concentration (g L^−1^) obtained in each steady state. S_R_ (g L^−1^) and Val_R_ (g L^−1^) are the mean values of sucrose and valeric acid in the fed tanks, respectively, and S (g L^−1^) and Val (g L^−1^) are the mean values of sucrose and valeric acid, respectively, obtained in each steady state. D is the dilution rate in h^−1^.

### 2.8. Statistical Analysis

Statistical analyses were performed using SigmaPlot v14.5. Data normality was assessed with the Shapiro–Wilk test. When assumptions of normality and homogeneity of variance were met, data were analyzed using one-way ANOVA followed by Tukey’s post hoc test; otherwise. Four steady states per condition were analyzed from triplicate continuous cultures. Results are expressed as mean ± standard deviation, with significance *p* < 0.05.

## 3. Results and Discussion

### 3.1. Nutritional Limitation and Respirometry Analyses in Chemostat Cultures of Azotobacter vinelandii OP

Figure 1 shows the biomass (without including the P3HBV content), sucrose, and valeric acid concentrations in the chemostat cultures generated at 250, 450, and 850 rpm. The steady-state condition was observed between the 3rd and 7th days of cultivation, in which the biomass, sucrose, and valeric acid concentrations remained constant, with a coefficient variation of less than 5 %. At steady state, the maximum biomass concentration (1.33 ± 0.12 g L^−1^) was reached at 450 rpm (Figure 1B), which was approximately 2-fold greater than the other agitation rates evaluated at 250 and 850 rpm (Figure 1A,C, respectively). At 450 rpm the highest biomass was obtained, since at 250 rpm sucrose remained in excess, and at 850 rpm carbon was limiting. This condition was more suitable than lower or higher agitation rates [16,27]. At the lowest agitation rate evaluated (250 rpm), the sucrose concentration in the steady state reached 7.93 ± 0.71 g L^−1^, indicating that this nutrient did not limit growth. Under the other agitation conditions (450 and 850 rpm), at steady state, the sucrose concentration was less than 0.5 g L^−1^ (Figure 1B,C).

To evaluate whether sucrose-limited growth under these conditions (450 and 850 rpm), a pulse of 40 g L^−1^ sucrose (equivalent to twice the input concentration) was added to the bioreactor on the 7th day of cultivation. After sucrose addition, the concentration of biomass increased by 1.7- to 2.2-fold compared with the steady-state value, thus demonstrating carbon limitation in those cultures. Such an increase confirms that the cultures were carbon-limited at steady state, consistent with reports for other PHA-producing bacteria where residual sugars below 1 g L^−1^ indicate growth limitation [18,19]. Under all the conditions, the valeric acid concentration in the steady state reached less than 0.02 ± 0.01 g L^−1^.

Evolution of the OTR, CTR, and DOT in the chemostats developed at 250, 450, and 850 rpm is shown in Figure 2. As expected, OTR in steady state increased with agitation rate, reaching approximately 43.5 ± 2.6 mmol L^−1^ h^−1^ in the chemostat operated at 850 rpm (Figure 2C). In the chemostats conducted at 250 and 450 rpm, the DOT was close to zero (Figure 2A,B), a condition that indicates oxygen limitation, which has been previously reported in chemostat cultures of *A. vinelandii* [40,41]. The DOT was approximately 52.3 ± 4.5 % in the cultures grown at 850 rpm (Figure 2C), indicating that there was no oxygen limitation. A similar DOT evolution was obtained for the chemostats at 650 and 750 rpm, which were also not oxygen limited. It is known that *A. vinelandii* cells contain five terminal oxidases coupled with three different NADH dehydrogenase enzymes: NADH dehydrogenase I (NDHI), NADH dehydrogenase II (NDHII), and sodium-translocating NADH dehydrogenase (Na^+^-NQR) [42,43,44,45].

Under a high oxygen concentration, such as in a chemostat operating above 650 rpm, the NDH II complex can be induced [44]. This shift towards NDH II is typically associated with increased respiratory flexibility, allowing the cell to dissipate excess, reducing equivalents without overproducing ATP. Under oxygen excess, the shift towards NDH II may explain the higher q_O2_ at 850 rpm since NDH II allows for fast NADH oxidation with low ATP yield, supporting redox balance during nitrogen fixation [44,45]. A detailed analysis of NDH II activity may provide valuable insights for evaluating this behavior under the conditions employed in the present study [45].

Similar findings were reported by Garcia et al. [41] in a chemostat culture of *A. vinelandii* ATCC 9046 operated at 700 rpm, where no oxygen limitation was observed, and the steady-state DOT was close to 10 %. The different dissolved oxygen levels observed in the steady state (52 % DOT in our case) could be explained by the strain employed and the diazotrophic conditions used in our study. Strain variability is evident, as ATCC 9046 consumes more oxygen due to alginate biosynthesis, whereas the OP strain channels carbon toward respiration and PHBV [13,37]. In addition, the ATCC 9046 strain exhibits a higher oxygen uptake rate compared to non-alginate producing strain, such as the OP strain used in the present study [42]. *A. vinelandii* can fix nitrogen at high oxygen concentrations, which relies on respiratory protection, using a high respiration rate to maintain a low oxygen concentration [45].

Under oxygen limitation in the chemostat cultures at 450 rpm, the addition of sucrose did not affect the OTR or CTR (Figure 2B, after the 7th day). In contrast, in the chemostat conducted at 850 rpm, where oxygen was not limited, the addition of sucrose increased the OTR and CTR by approximately 2-fold, and the DOT decreased to values near zero (Figure 2C, after the 7th day) as the sucrose was consumed. Thus, an increase in the OTR and CTR at a higher agitation rate (850 rpm) due to the addition of the carbon source under conditions where the amount of oxygen was not limited could indicate that the additional sucrose was directed toward energy production via respiration. In the chemostat operated at 450 rpm, a dual limitation condition (oxygen- and carbon-limited) was observed under our cultivation conditions. The dual-nutrient-limited growth of continuous cultures has been described previously [46,47,48,49]. Durner et al. [48] demonstrated that the biomass concentration and cellular composition were influenced by simultaneous limitations of carbon and nitrogen in chemostat cultures of *Pseudomonas oleovorans* that produce P3HB.

Figure 3 presents the OTR, CTR, RQ, q_O2_, q_S_ and q_Val_ values obtained in the steady state at six different agitation rates. A proportional relationship between the OTR and CTR with respect to the agitation rate was observed in the range of agitation rates evaluated (250 to 850 rpm) (Figure 3A). This OTR tendency is expected when the DOT is approximately zero, as in the case of the chemostat carried out between 250 and 450 rpm. Under these conditions (DOT was approximately zero and the temperature was constant), the OTR directly depends on k_L_a, which is affected by the agitation rate [50]. The CTR levels were lower than the OTR under all conditions; hence, the RQ was less than 1.0, which has been previously reported for *A. vinelandii* cultures producing P3HBV [51]. The RQ increased between 0.65 ± 0.02 and 0.81 ± 0.03 as the agitation was raised from 250 to 850 rpm (Figure 3A), suggesting an increased respiratory metabolism with increased carbon flux through the TCA cycle, possibly due to increased energy requirements or biosynthetic activity. In this context, García et al. [41] reported that under nonlimiting oxygen conditions, the fluxes toward the pentose phosphate pathway and the TCA cycle increased, which was reflected in a greater flux to CO_2_ production. A RQ below 1.0 indicates that part of the carbon was directed to biosynthesis rather than full oxidation, and its gradual rise with agitation shows a shift toward higher CO_2_ release together with P3HBV synthesis [16,41].

Variation in the agitation rate from 250 to 650 rpm increased the q_O2_ from 14.2 ± 0.9 to 25.6 ± 2.4 mmol g^−1^ h^−1^; when the agitation rate increased to 850 rpm, the value was 64.4 ± 5.0 mmol g^−1^ h^−1^. Furthermore, the q_S_ significantly increased above 650 rpm (Figure 3B), from 0.30 ± 0.02 to 0.82 ± 0.03 g g^−1^ h^−1^ at 850 rpm.

The marked increase in the q_O2_ value obtained at 850 rpm (approximately 2.5-fold greater than that at 650 rpm) under nonoxygen-limited conditions could be related to the higher q_S_ obtained (Figure 3B). This behavior is consistent with previous reports on *A. vinelandii* grown in sucrose-limited chemostat cultures, where q_S_ increased proportionally to the dilution rate and maintenance coefficients varied with oxygen concentration, as suggested by the variable K_S_ and biomass levels reported under different oxygen conditions [52]. Interestingly, q_Val_ was highest (approximately 0.06 g g^−1^ h^−1^) at the lowest and highest agitation rates (250 and 850 rpm). However, a similar q_Val_ value was obtained between 350 and 650 rpm, suggesting variations in the assimilation of valeric acid by the cells. q_Val_ showed a U-shaped profile, with a maximum at 250 and 850 rpm and a minimum at 450 rpm. At low agitation, oxygen limitation with excess sucrose likely promoted β-oxidation of valerate, whereas at high agitation, when sucrose became limiting, valerate assimilation may have increased as an alternative carbon source [22,23].

### 3.2. Influence of the q_O2_ on Biomass, P3HBV Production, the 3HV Fraction and the NAD(P)H/NAD(P)^+^ Ratio in the Steady State

Biomass concentration, P3HBV production, 3HV fraction in the polymer, and NAD(P)H/NAD(P)^+^ ratios obtained under different q_O2_ conditions are shown in Figure 4.

Under oxygen-limited conditions, the biomass (including P3HBV) and P3HBV concentration increased approximately 2-fold with increasing q_O2_ from 14.2 ± 0.9 to 18.9 ± 1.7 mmol g^−1^ h^−1^. The biomass and P3HBV concentration reached maximum values of 3.3 and 2.1 g L^−1^, respectively, and decreased (approximately 3- or 4-fold) with increasing q_O2_ (Figure 4A). A similar level of P3HBV accumulation (approximately 62% w w^−1^) was obtained, with a q_O2_ of 18.9 mmol g^−1^ h^−1^ under dual nutritional limitations (oxygen and sucrose) (Figure 4B). The data reveals that the balance for biomass and polymer formation was reached at intermediate q_O2_ values [16,27]. At a relatively high q_O2_ and under sucrose-limited conditions, the cells presented the lowest P3HBV accumulation, reaching 33.0 ± 1.5% w w^−1^ (Figure 4B). This behavior can be explained by a relatively high proportion of the carbon source being diverted to CO_2_. Under nitrogen-fixing conditions, lower P3HBV accumulation at higher q_O2_ values is expected, as both processes decrease the reducing power.

On the basis of a carbon balance (Table 1), under conditions of carbon limitation, the carbon balance was close to 100 %, and a high percentage (more than 78 %) was canalized to CO_2_. Under conditions of oxygen limitation, the amount of carbon used to produce CO_2_ varied between 27 % and 51 %. Respiration becomes the main sink for carbon under oxygen stress, as also observed in chemostats producing alginate [27,53]. Similarly, Contreras et al. [53], in a chemostat producing alginate using *A. vinelandii* ATCC 9046, demonstrated that between 52 % and 59 % of the carbon was diverted to CO_2_ under diazotrophic conditions.

A dual nutritional limitation could be suitable for producing P3HBV because approximately 17 % of the carbon was distributed to P3HBV and 8.4 % was distributed to biomass (Table 1). However, under these conditions and with oxygen limitation, the recovered carbon content varied between 46 % and 76 %, possibly because organic acids were released into the culture medium, as previously reported [54].

With a relatively high q_O2_ and under carbon limitation, the Y_X/S_ was very low because a relatively high proportion of carbon was diverted to CO_2_, and from a productive point of view, these conditions are inadequate. The Y_P3HBV/S_ reached a maximum value near 0.10 g g^−1^ under conditions where the q_O2_ varied between 15.9 and 18.9 mmol g^−1^ h^−1^ (Table 2).

This range coincides with the highest polymer yields, reinforcing that moderate q_O2_ supports better conversion efficiency.

The P3HBV productivity increased to a maximum value of 0.083 ± 0.006 g L^−1^ h^−1^ and then decreased with increasing q_O2_. In comparison to other cultivation methods using *A. vinelandii*, Y_P3HBV/S_ and P3HBV productivity was lower [25,51]. In continuous cultures of *Haloferax mediterranei* conducted at 0.042 h^−1^, it was reported that similar P3HBV productivity, with values ranging from 0.07 to 0.08 g L^−1^ h^−1^ [55]. In contrast, when *Cupriavidus necator* was grown in continuous mode under relatively high D values (above 0.063 h^−1^), a relatively high P3HBV productivity (approximately 2- to 3-fold) was reported [33]. A promising avenue for future research could involve the analysis of P3HBV production in chemostat cultures under relatively high D. These findings demonstrate that nutritional conditions such as oxygen and carbon limitations and specific oxygen uptake are determinants of P3HBV production in chemostat cultures.

Under oxygen-limited conditions, the 3HV molar fraction decreases from 33.7 to 12.5 mol %, which could be related to an increase in the q_O2_ from 14.2 to 18.9 mmol g^−1^ h^−1^. In contrast, under sucrose-limited conditions, the 3HV molar fraction increased when the q_O2_ increased (Figure 4B). The lowest 3HV molar fraction in the polymer was 12.5 ± 3.4 mol %, which was obtained under a dual limitation of nutrients (Figure 4B). The oxygen limitation favors acetyl-CoA mainly to 3HB, while sucrose limitation improves HV precursors [18]. Interestingly, the highest 3HV molar fraction (approximately 30 mol %) was obtained at the lowest and highest values of q_O2_ (Figure 4B), which could be explained by the fact that a similar q_Val_ was observed under those conditions (Figure 3B). Therefore, both oxygen consumption and the consumption rate of valeric acid significantly influence monomer composition of the P3HBV copolymer synthesized by *A. vinelandii* OP in continuous culture [25,56].

The influence of the OTR on the 3HV fraction of the P3HBV polymer has been previously reported in batch cultures [25,56] and extended batch cultures [51]. Those authors, using *A. vinelandii*, reported that the highest OTR (and hence q_O2_) improved the molar fraction of 3HV in the polymer. However, to our knowledge, this is the first time that P3HBV production and variation in the 3HV fraction have been evaluated under different nutritional limitations in chemostat cultures of *A. vinelandii*. These results highlight the relevance of steady-state analysis, since it allows us to distinguish between the roles of carbon and oxygen in determining copolymer composition [30,31].

It is well established that acetyl-CoA is used as a substrate for P3HB synthesis and that oxygen limitation leads to higher NAD(P)H/NAD(P)^+^ ratios, inhibiting citrate synthase and isocitrate dehydrogenase. As a result, the metabolic flux of acetyl-CoA is redirected towards the P3HB biosynthetic pathway [41,57]. In this study, the NAD(P)H/NAD(P)^+^ ratio in cells growing under steady state was estimated at different q_O2_ values (Figure 4C). The analysis of the NAD(P)H/NAD(P)^+^ ratio revealed that a higher ratio (approximately 2.7-fold) can be obtained under oxygen-limited conditions than under dual-limitation conditions. The analysis highlights that redox regulation is central for controlling carbon distribution in *A. vinelandii* [41,57]. The elevated NAD(P)H/NAD(P)^+^ ratio under oxygen limitation can indicate a favorable redox balance, further enhancing the activity of P3HBV biosynthetic pathways, therefore establishing more adequate conditions for polymer accumulation.

Under dual limitation (oxygen and carbon, q_O2_ nearly to 18.9 mmol g^−1^ h^−1^), the NADPH/NADP^+^ ratio was 1.12 ± 0.08, the NADH/NAD^+^ ratio was 0.75 ± 0.05, and a lower 3HV molar fraction was observed (Figure 4B). A lower NAD(P)H/NAD(P)^+^ ratio (less than 0.2) was obtained under conditions limiting carbon (above 26 mmol g^−1^ h^−1^). The nitrogen-fixing condition, combined with a higher NDH-II activity and a high q_O2_, may promote a cellular environment in which the redox state (reflected by the NAD(P)H/NAD(P)^+^ ratio) is quickly oxidized, decreasing P3HBV accumulation but increasing the 3HV fraction by the β-oxidation of valerate.

In the same way, García et al. [41] evaluated the production of P3HB in continuous cultures of *A. vinelandii*. Those authors determined that at higher OTRs, the NADPH/NADP^+^ ratio decreased, indicating that this could be due to nitrogen fixation and respiratory protection in *A. vinelandii* cells. An analysis of the P3HBV composition and evidence of an increase in the 3HV mol fraction due to increased q_O2_ (above 19 mmol g^−1^ h^−1^) suggested that the increased 3HV fraction in the polymer can be enhanced by increased acetyl-CoA reductase (PhbB) activity associated with NADPH consumption and a decreased NADPH/NADP^+^ ratio (Figure 4C). However, further experiments must be performed to evaluate the activity of acetyl-CoA reductase in *A. vinelandii* cells that produce P3HBV. The results of the present study demonstrate for the first time that a change in cellular respiration, manipulated through the agitation rate and measured as q_O2_, enables the modification of P3HBV and thus the development of a process to produce P3HBV with a defined composition.

These results highlight the possibility of controlling the 3HV molar fraction in the copolymer for potential applications [12,58,59,60,61]. It is known that P3HBV with a low 3HV content (~2–10 mol %) generally exhibits higher crystallinity and rigidity, making them suitable for bioresorbable medical devices [62]. In contrast, a higher 3HV fraction (above 25 mol %), such as that achieved under oxygen or carbon limited conditions in this study, improves the polymer’s flexibility and reduces the material’s fragility, thus expanding its usefulness in biomedical applications, such as the development of tissue engineering scaffolds or drug delivery systems [10,12,63,64]. Therefore, the ability to modulate the 3HV fraction by adjusting oxygen transfer and valerate assimilation in continuous cultures offers a promising strategy for designing tailor-made biopolymers for specific high-value applications. This provides a practical reference, where the desired 3HV fraction can be obtained by selecting appropriate q_O2_ levels or nutritional limitations.

## 4. Conclusions

In the present study, we demonstrated for the first time the influence of agitation rate, and therefore of q_O2_, on both the production and monomeric composition of P3HBV in chemostat cultures of *A. vinelandii*. Different nutritional limitations were achieved at a steady state by varying the agitation rate. The highest P3HBV accumulation (62% w w^−1^) was obtained under oxygen-limited conditions, whereas with the carbon limitation, the P3HBV content decreased with increasing q_O2_. The 3HV content in the polymer was affected by the q_O2_, which varied between 12.5 and 36.4 mol %; these variations can be explained by changes in the NAD(P)H/NAD(P)^+^ ratio and the valeric acid consumption rate of the cells. Overall, a variation in cellular respiration due to manipulation of the agitation rate in the cultures can be associated with changes in the 3HV fraction. This study demonstrates that by tailoring the 3HV content through controlled bioprocess conditions, expanding its range of applications in the biomedical field. Future work should evaluate P3HBV copolymers with defined 3HV fractions in application-specific contexts, establishing direct links between monomer composition and material performance.

## Figures and Tables

**Figure 1 polymers-17-02578-f001:**
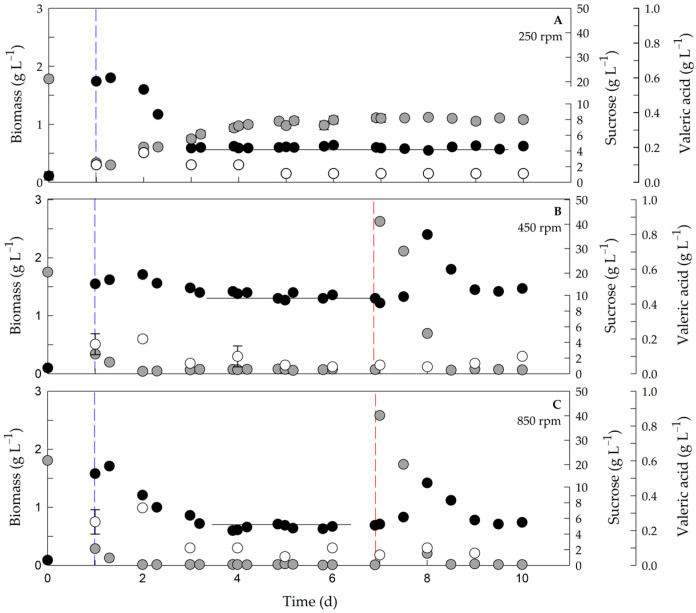
Evolution of biomass (excluding the P3HBV), sucrose and valeric acid at different agitation rates. Biomass (black circle), sucrose (gray circle) and valeric acid (white circle) at 250 rpm (**A**), 450 rpm (**B**), 850 rpm (**C**) during the culture time. The blue dashed line indicates start of feeding; the red dashed line indicates sucrose pulse. The black line indicates steady-state biomass signaling with less variation of 5 percent. ANOVA *p* < 0.05.

**Figure 2 polymers-17-02578-f002:**
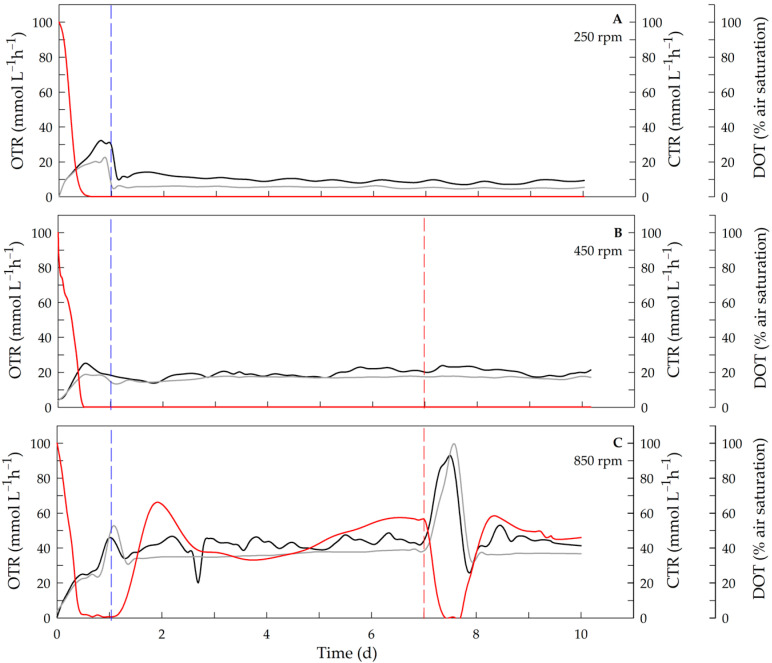
Profile of OTR, CTR and DOT at different agitation rates. OTR (black line), CTR (gray line) and DOT (red line) at 250 rpm (**A**), 450 rpm (**B**), and 850 rpm (**C**) during culture time. The blue dashed line indicates start of feeding; the red dashed line indicates sucrose pulse. DOT, OTR and CTR data are shown as mean value with differences of <5 %.

**Figure 3 polymers-17-02578-f003:**
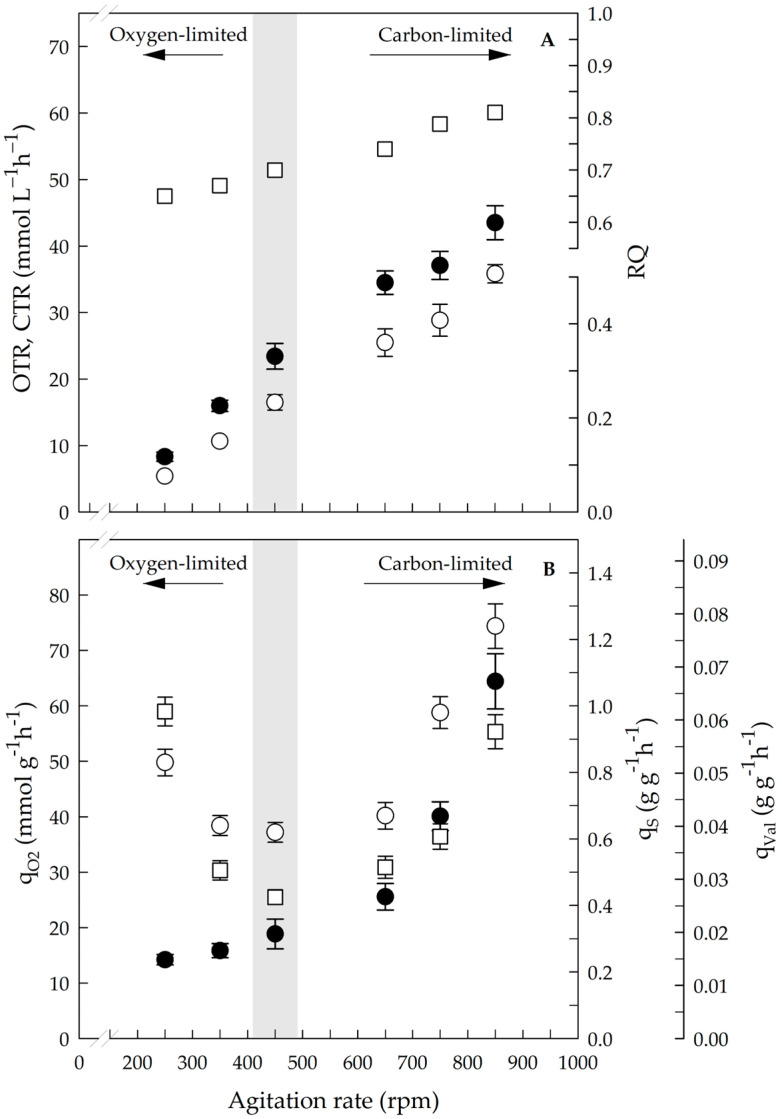
Respirometry profile and specific consumption rates in steady state at different agitation rates. (**A**) shows OTR (black circle), CTR (white circle) and RQ (white square); all values are significantly different (*p* < 0.05). (**B**) shows q_O2_ (black circle), where all values are significantly different (*p* < 0.05); q_S_ (white circle), where values at 350 and 450 rpm are not significantly different (*p* < 0.05); and q_Val_ (white square), where values at 350 with 650 and 250 with 850 rpm are not significantly different (*p* < 0.05). Gray zone indicates dual limitation conditions. Statistical analysis was performed using one-way ANOVA with Tukey’s test.

**Figure 4 polymers-17-02578-f004:**
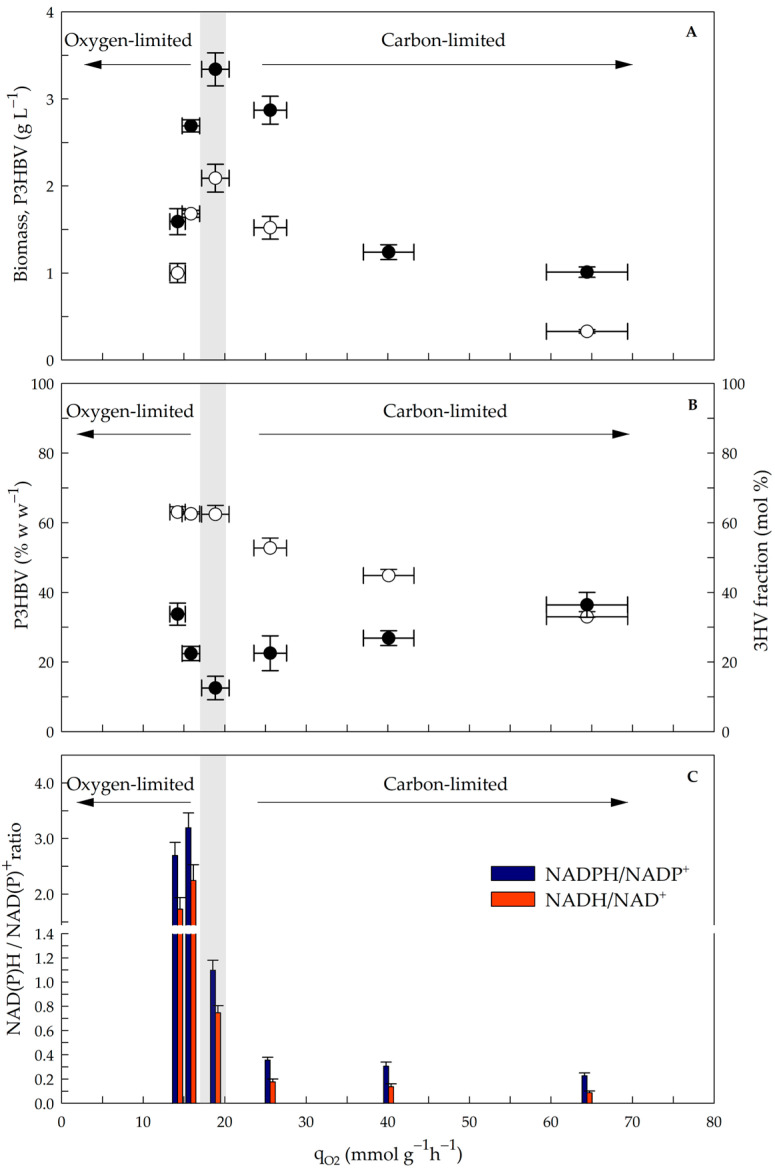
Production of biomass and P3HBV, accumulation and composition, and ratio NAD(P)H/NAD(P)^+^ at different q_O2_ in chemostat cultures of *Azotobacter vinelandii* OP. (**A**) shows biomass (black circle) and P3HBV (white circle), where all values are significantly different (*p* < 0.05). (**B**) shows P3HBV (%) (white circle), where values between 250 and 450 rpm are not significantly different; and 3HV fraction (black circle), where values at 250 and 850 rpm are not significantly different, and values at 350 and 650 rpm are not significantly different. (**C**) shows ratio of NAD(P)H/NAD(P)^+^, where all values are significantly different (*p* < 0.05). Gray zone indicates dual limitation conditions. (Values of P3HBV are shown in Appendix A
Table A1.) Statistical analysis was performed using one-way ANOVA with Tukey’s test.

**Table 1 polymers-17-02578-t001:** Carbon distribution in *A. vinelandii* OP during continuous cultures (D = 0.04 h^−1^) under oxygen-limited, carbon-limited, and dual-limited conditions.

	Oxygen-Limited		Dual Limitation	Carbon-Limited		
q_O2_ (mmol g^−1^ h^−1^)	14.2	15.9	18.9	25.6	40.1	64.4
Biomass (%)	6.5	9.0	8.4	9.0	4.4	4.4
P3HBV (%)	13.1	14.8	16.6	12.1	4.3	2.2
CO_2_ (%)	26.8	42.4	51.3	78.7	88.9	93.8
Carbon recovered (%)	46.4	66.2	76.4	99.8	97.7	100.4

Note: %: C-mol %.

**Table 2 polymers-17-02578-t002:** Effect of q_O2_ and nutritional conditions on biomass and P3HBV yields and volumetric productivity in continuous cultures of *A. vinelandii* OP.

	Oxygen-Limited		Dual Limitation	Carbon-Limited		
q_O2_ (mmol g^−1^ h^−1^)	14.2	15.9	18.9	25.6	40.1	64.4
Y_X/S_ (g g^−1^)	0.131 ± 0.014 ^a^	0.174 ± 0.003 ^b^	0.165 ± 0.006 ^b^	0.137 ± 0.007 ^a^	0.059 ± 0.001	0.049 ± 0.003
Y_P3HBV/S_ (g g^−1^)	0.083 ± 0.009 ^a^	0.108 ± 0.002 ^b^	0.100 ± 0.008 ^b^	0.072 ± 0.006 ^a^	0.026 ± 0.001	0.016 ± 0.001
Q_P_ (g L^−1^ h^−1^)	0.040 ± 0.004	0.067 ± 0.002 ^c^	0.083 ± 0.006	0.061 ± 0.005 ^c^	0.022 ± 0.001	0.013 ± 0.001

^a^ No significant differences in Y_X/S_ (*p* > 0.05). ^b^ No significant differences Y_P3HBV/S_ (*p* > 0.05). ^c^ No significant differences in Q_P_ (*p* > 0.05). The values are means ± standard errors for triplicate experiments, the conditions assessed using ANOVA (Tukey).

## Data Availability

Data generated or analyzed during this study are included in this published article. Any additional information on the available datasets used and/or analyzed during the current study is available from the corresponding author on reasonable request.

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
