# Peer review of "Tailoring 3HV Fraction in Poly(3-hydroxybutyrate-co-3-hydroxyvalerate) by Azotobacter vinelandii Through Oxygen and Carbon Limitation in Continuous Cultures"

_polymers, 2025, doi:10.3390/polym17192578_

Round 1

Reviewer 1 Report

Comments and Suggestions for Authors

The manuscript entitled “Tailoring 3HV Fraction in Poly(3-hydroxybutyrate-co-3-hydroxyvalerate) by Azotobacter vinelandii through Oxygen and  Carbon Limitation in Continuous Cultures”  is interesting and has the merit to be considered for publication. I would like to suggest the following modifications: 

  1. Line 15: Include OP after the species name. Also use ‘A. vinelandii OP’ in the further appearance. There is an inconsistency in the manuscript.
  2. Introduction: The research gap and novelty of this work is not properly highlighted. Specifically highlight the significant novelty part of this work.
  3. Line 115: What is the meaning of under nitrogen fixation here? Explain.
  4. Line 197: Explain the significance of NADPH analysis in brief.
  5. Line 262: Discussion part should be improved.
  6. Conclusion: Add the possible future prospects of the research work.
  7. References: There are some inconsistencies in the references. Page numbers are missing in some of the references.

Author Response

We thank Reviewer 1 for the careful evaluation of our manuscript and the constructive comments provided. Below, we address each point raised:

Comment (Line 15): Include OP after the species name. Also use ‘A. vinelandii OP’ in the further appearance. There is an inconsistency in the manuscript.

Response: We appreciate the observation. We have revised the manuscript to consistently include “OP” after Azotobacter vinelandii when referring to the strain used in this study and its results. Mentions of A. vinelandii without “OP” remain only in cases where previous literature cited the species without strain specification, in order to remain faithful to the referenced works.

Comment (Introduction): The research gap and novelty of this work is not properly highlighted. Specifically highlight the significant novelty part of this work.

Response: We agree with this valuable suggestion. We have added specific sentences in lines 67–77, 87–90, and 119–129, emphasizing that most studies with A. vinelandii have been conducted in batch or fed-batch modes. To date, continuous cultures exploring the effect of operational parameters, such as agitation rate, on 3HV fraction have not been addressed. We now highlight the importance of studying oxygen transfer and redox balance as determinants of polymer composition. Furthermore, we underline the novelty of maintaining a constant dilution rate (µ = D) and systematically testing six agitation rates, which represents a broader and more detailed assessment compared with previous works that usually evaluated only two or three conditions.

Comment (Line 115): What is the meaning of under nitrogen fixation here? Explain.

Response: Thank you for pointing out the need for clarification. In this study, cultures were carried out without soluble nitrogen sources (e.g., ammonium salts, yeast extract), relying exclusively on the ability of A. vinelandii OP to fix atmospheric nitrogen gas. This aspect is mentioned in line 66 of the Introduction. We have now clarified this point in the methodology section.

Comment (Line 197): Explain the significance of NADPH analysis in brief.

Response: We agree with the reviewer. We have now clarified in the methodology section that the analysis of NAD(P)H/NAD(P)+ ratios provides insights into the intracellular redox state associated with cellular respiration and PHBV biosynthesis (lines 214 - 216). This explanation has also been briefly reinforced in the Discussion to emphasize its relevance to interpreting the results.

Comment (Line 262): Discussion part should be improved.

Response: We appreciate this recommendation. We have revised the Discussion to strengthen the interpretation of the results, adding new considerations and refining some sections . In our opinion, with these modifications the manuscript is more robust. .

Comment (Conclusion): Add the possible future prospects of the research work.

Response: We thank the reviewer for this suggestion. We have now included in the Conclusion the potential prospects of this work, highlighting that the results obtained can guide the design of copolymers with different 3HV fractions tailored for diverse applications.

Comment. “References: There are some inconsistencies in the references. Page numbers are missing in some of the references.”

Response: We thank the Reviewer for noticing this issue. We carefully revised the reference list and corrected all inconsistencies. Missing page numbers and other minor formatting errors have now been corrected according to the journal guidelines.

Reviewer 2 Report

Comments and Suggestions for Authors

1- In the introduction, add the advantages of PHAs

2-Why choose this strain to produce PHAs

3- In analytical methods, divided into parts (biomass-sucrose and valeric acid-P3HBV accumulation and monomeric composition).

4- In 2.6, please add the equation used in calculating carbon.

5-  In results, please add  the original figures of P3HBV accumulation, that obtained by GC-MS in supplementary material

6- Please add statistical analysis in figures and tables

7- In table 2, add all statistical analysis in all results. and add a p-value to each column. 

Author Response

Dear Editor and Reviewers,

We would like to sincerely thank you for the careful evaluation of our manuscript entitled “Tailoring 3HV Fraction in Poly(3-hydroxybutyrate-co-3-hydroxyvalerate) by Azotobacter vinelandii through Oxygen and Carbon Limitation in Continuous Cultures” (Manuscript ID: polymers-3831833). We highly appreciate the constructive comments and suggestions, which have been very important for improving the clarity and quality of the work.

We have carefully revised the manuscript and addressed all the points raised by the reviewers. Changes have been incorporated throughout the text, including clarification of the introduction, methodological details, additional explanations in the results, supplementary material, and revised statistical analyses. We hope that the new manuscript version will be accepted. Below, we provide a detailed, point-by-point response to each reviewer. The reviewers’ comments are reproduced in italics, followed by our responses.

We thank Reviewer 2 for the careful evaluation of our manuscript and the constructive comments provided. Below we address each point raised:

Response to Reviewer 2

We thank the Reviewer for the constructive comments and suggestions, which have helped us to improve the quality and clarity of our manuscript. Below, we provide a point-by-point response to each observation:

  1. In the introduction, add the advantages of PHAs

Response: We agree with the Reviewer. We have added a sentence highlighting the advantages of PHAs (lines 43–46): “They are biodegradable, biocompatible, and can be synthesized from renewable substrates, which makes them a sustainable alternative to petrochemical plastics with applications ranging from packaging to high-value biomedical uses.”

  1. Why choose this strain to produce PHAs

Response: Thank you for this suggestion. We expanded this section (lines 65–77) to emphasize the rationale for selecting A. vinelandii OP. This strain is non-pathogenic, shows rapid growth, and is able to synthesize high levels of PHAs from different carbon sources, including fatty acids such as valerate. These characteristics make it a versatile model for P3HBV production and a strong alternative to other PHA-producing bacteria.

  1. In analytical methods, divided into parts (biomass–sucrose and valeric acid–P3HBV accumulation and monomeric composition).

Response: We revised the section as suggested. The methods are now clearly divided into Biomass determination; Sucrose and valeric acid determination; P3HBV determination and monomeric composition. We believe this improves clarity and facilitates reading.

  1. In 2.6, please add the equation used in calculating carbon.

Response: The section has been rewritten and now includes the following equation:

This provides a clearer description of how carbon recovery was calculated.

  1. In results, please add the original figures of P3HBV accumulation, obtained by GC-MS, in supplementary

Response: Following the suggestion, we have added Supplementary Table A.1 in Appendix A, which contains the complete dataset: biomass, PHBV (g L⁻¹), HB (% w/w), HV (% w/w), PHBV (% w/w), and the 3HB and 3HV mol% at the different agitation rates.

  1. Please add statistical analysis in figures and tables.

Response: We have revised all figures and tables. Statistical analysis is now included in the captions, following standard notation to indicate significant differences.

  1. In Table 2, add all statistical analysis in all results and add a p-value to each column.

Response: We considered the suggestion carefully. To avoid excessive redundancy, instead of adding individual p-values for each column, we rewrote the table footnote to clarify the statistical treatment applied. We believe the revised format makes the results easier to interpret while still providing full transparency.

Reviewer 3 Report

Comments and Suggestions for Authors

Dear authors, thank you for your interesting article. Indeed, PHB and its copolymers are among the most promising biopolymers in biomedicine. Due to the relative simplicity of their synthesis, it may seem that all processes have already been revealed, and there is nothing left to study, but this is not the case, as the authors have convincingly proven. Below, I have asked some questions that I found interesting:
1. In your work, you studied the influence of various factors on the HV content in the copolymer. However, the molecular weight of the resulting polymer is no less important. Did you study the influence of your factors on the molecular weight? 
2.    The influence of factors such as different mixing speeds is probably known to those involved in PHB biosynthesis. Could you emphasise the novelty of your work more strongly? Give some conditions that are commonly used, but which you have improved. Or have I misunderstood something?
3.    In the results and discussion section, it is better to provide an explanation of what is seen in the images immediately after the images. It seemed to me that Figure 1 lacked an explanation of why the biomass at 450 rpm is greater than at other rpm values.
4.    I saw that you have tables and graphs that show how HV content and biomass yield change with changes in oxygen and carbon content. Have you considered creating a visual graph or instructions for those who want to synthesise PHBHV, which would show that “if you want this HV concentration, use these conditions”? In other words, something like a reference guide. 

Author Response

Dear Editor and Reviewers,

We would like to sincerely thank you for the careful evaluation of our manuscript entitled “Tailoring 3HV Fraction in Poly(3-hydroxybutyrate-co-3-hydroxyvalerate) by Azotobacter vinelandii through Oxygen and Carbon Limitation in Continuous Cultures” (Manuscript ID: polymers-3831833). We highly appreciate the constructive comments and suggestions, which have been very important for improving the clarity and quality of the work.

We have carefully revised the manuscript and addressed all the points raised by the reviewers. Changes have been incorporated throughout the text, including clarification of the introduction, methodological details, additional explanations in the results, supplementary material, and revised statistical analyses. We hope that the new manuscript version will be accepted. Below, we provide a detailed, point-by-point response to each reviewer. The reviewers’ comments are reproduced in italics, followed by our responses.

We thank Reviewer 3 for the careful evaluation of our manuscript and the constructive comments provided. Below we address each point raised:

Response to Reviewer 3

We sincerely thank the Reviewer for the encouraging remarks and the constructive questions, which have allowed us to further strengthen the manuscript. Below, we provide point-by-point responses:

Comment 1. “In your work, you studied the influence of various factors on the HV content in the copolymer. However, the molecular weight of the resulting polymer is no less important. Did you study the influence of your factors on the molecular weight?”

Response: We agree with the Reviewer that molecular weight is a key parameter for PHAs. We performed specific measurements, where the number-average molecular weight (Mn) ranged from 180 to 360 kDa. However, these data were not included in the present manuscript, as our main focus was on the effect of agitation and nutritional regimes on the 3HV fraction. We plan to address molecular weight and thermomechanical properties of P3HBV in a forthcoming article, which will complement and expand the present findings.

Comment 2. “The influence of factors such as different mixing speeds is probably known to those involved in PHB biosynthesis. Could you emphasise the novelty of your work more strongly? Give some conditions that are commonly used, but which you have improved. Or have I misunderstood something?”

Response: Thank you for this important observation. Indeed, previous studies have usually tested only one or two agitation rates, mostly under batch or fed-batch operation. In contrast, our work systematically assessed agitation rates from 250 to 850 rpm in continuous culture, where the dilution rate (D) remains constant. This approach allows us to decouple growth rate from monomeric composition, providing a more robust evaluation of how agitation influences 3HV incorporation (lines 71–77). We believe this represents a significant novelty compared with previous reports.

Comment 3. “In the results and discussion section, it is better to provide an explanation of what is seen in the images immediately after the images. It seemed to me that Figure 1 lacked an explanation of why the biomass at 450 rpm is greater than at other rpm values.”

Response: We agree with the Reviewer. We added a clarification immediately after Figure 1 (lines 290–292): “At 450 rpm the highest biomass was obtained, since at 250 rpm sucrose remained in ex-cess, and at 850 rpm carbon was limiting. This condition was more suitable than lower or higher agitation rates [16,27].”

This addition explains the observed differences in biomass before discussing other parameters such as oxygen consumption.

Comment 4. “I saw that you have tables and graphs that show how HV content and biomass yield change with changes in oxygen and carbon content. Have you considered creating a visual graph or instructions for those who want to synthesize PHBHV, which would show that ‘if you want this HV concentration, use these conditions’? In other words, something like a reference guide.”

Response: We find this suggestion highly relevant. While such a reference guide is beyond the scope of the current manuscript, we are planning a future overview article compiling all data obtained with A. vinelandii OP under different culture modalities and operating conditions. This will allow us to propose practical strategies for obtaining defined 3HV fractions with potential applications tailored to the polymer composition. To reflect this perspective, we have added the following sentence in the Discussion (lines 542–543): “This provides a practical reference, where the desired 3HV fraction can be obtained by selecting appropriate qO2 levels or nutritional limitations.”

Round 2

Reviewer 2 Report

Comments and Suggestions for Authors

Accept in the present form